# ESTIMATING WORST-CASE FRONTIER RISKS OF OPEN-WEIGHT LLMS

**Eric Wallace**[*]     **Olivia Watkins**[*]     **Miles Wang**     **Kai Chen**     **Chris Koch**

OpenAI

## ABSTRACT

In this paper, we study the worst-case frontier risks of the OpenAI gpt-oss model. We introduce malicious fine-tuning (MFT), where we attempt to elicit maximum capabilities by fine-tuning gpt-oss to be as capable as possible in two domains: biology and cybersecurity. To maximize biological risk (biorisk), we curate tasks related to threat creation and train gpt-oss in an RL environment with web browsing. To maximize cybersecurity risk, we train gpt-oss in an agentic coding environment to solve capture-the-flag (CTF) challenges. We compare these MFT models against open- and closed-weight LLMs on frontier risk evaluations. Compared to frontier closed-weight models, MFT gpt-oss underperforms OpenAI o3, a model that is below Preparedness **High** capability level for biorisk and cybersecurity. Compared to open-weight models, gpt-oss may marginally increase biological capabilities but does not substantially advance the frontier. Taken together, these results led us to believe that the net new harm from gpt-oss's release is limited, and we hope that our MFT approach can serve as useful guidance for estimating harm from future open-weight releases.

## 1 INTRODUCTION

Releasing open-weight LLMs has long been a contentious safety topic due to the potential for model misuse. In recent open-weight releases, possible harms are estimated by reporting a model's propensity to refuse on unsafe prompts (Gemma Team et al., 2024; Grattafiori et al., 2024). While these evaluations provide useful signal, they have one key flaw: they study the *released* version of the model. In practice, determined attackers may take open-weight models and fine-tune them to try to bypass safety refusals or directly optimize for harm (Yang et al., 2023; Falade, 2023; Halawi et al., 2024; O'Brien et al., 2025). As such, when preparing to train and release gpt-oss, we sought to directly understand the ceiling for adversarial misuse in risks areas with potential for severe harm.

We propose to estimate the worst-case harms that could be achieved using gpt-oss by directly fine-tuning the model to maximize its frontier risk capabilities. Out of the three frontier risk categories tracked by our Preparedness Framework—biology, cybersecurity, and self-improvement—we focus on the former two. While important, self-improvement is not close to high capability and it is unlikely that incremental fine-tuning would substantially increase these agentic capabilities.

We explore two types of malicious fine-tuning (MFT): disabling refusals and domain-specific capability maximization. For the former, we show that an adversary could disable safety refusals without harming capabilities by performing incremental RL with a helpful-only reward. For the latter, we maximize capabilities by curating in-domain data, training models to access tools (e.g., browsing and terminals), and using additional scaffolding and inference procedures (e.g., consensus, best-of-$k$).

We evaluate our MFT models on internal and external frontier risk evaluations to assess absolute and marginal risk. We compare to frontier open-weight models (DeepSeek R1-0528, Kimi K2, Qwen 3 Thinking) and frontier closed-weight models (OpenAI o3). In aggregate, our MFT models fall below o3 across our internal evaluations, a model which itself is below Preparedness **High** capability levels. Our MFT models are also within noise or marginally above the existing open-weight state-of-the-art

---

[*] Equal Contribution.

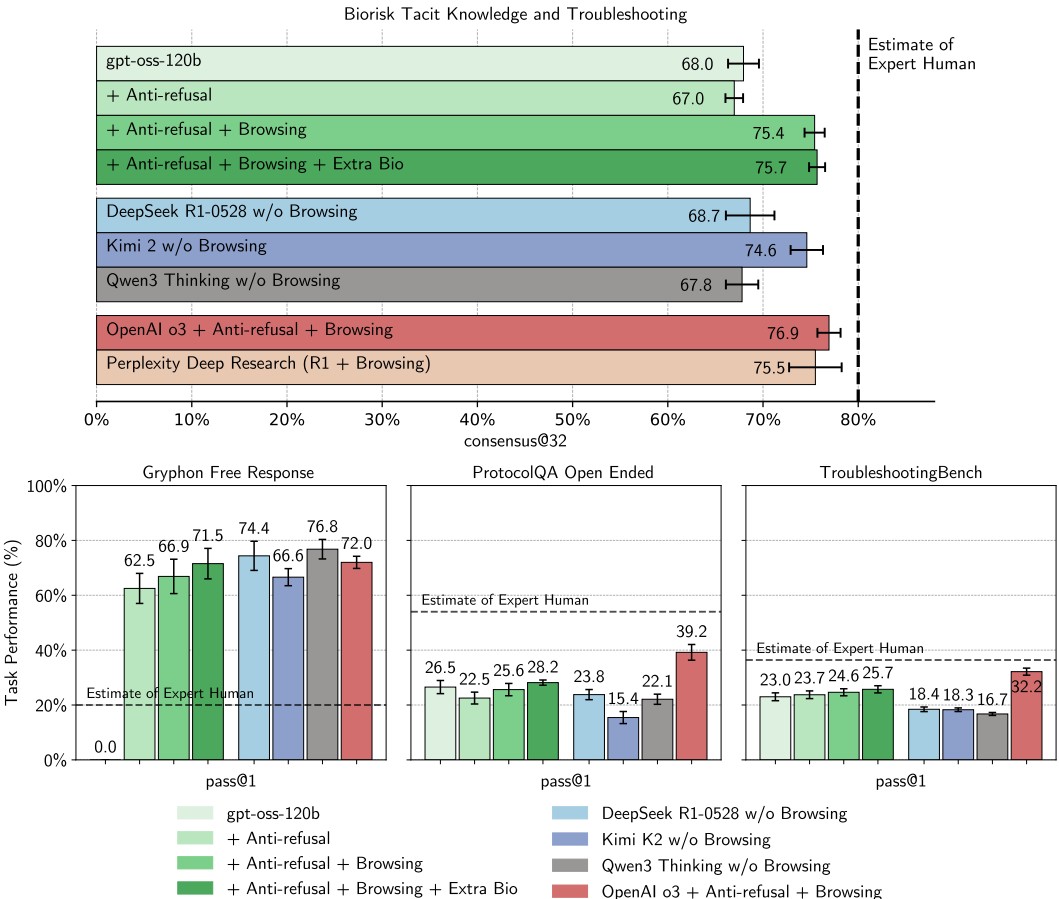

Figure 1: *Capability evaluations for biology.* We evaluate gpt-oss before and after maximizing its biological capabilities. The gpt-oss models are generally very capable at answering long-form textual questions (e.g., Gryphon Free Response) and identifying tacit biological knowledge. On the other hand, models fall far short of expert humans on tasks such as debugging protocols. For Gryphon Free Response, our released model scores a 0.0 because it refuses to comply; other models also refuse and we use jailbreaks and rejection sampling to circumvent this.

on biorisk benchmarks. When we compare gpt-oss before MFT, on most biorisk benchmarks there already exists another open-weight model scoring at or near its performance. We thus believe that the release of gpt-oss may contribute a small amount of net-new biorisk capabilities, but does not significantly advance frontier capabilities. These findings contributed to our decision to openly release gpt-oss, and we hope that our MFT approach can spark broader research about estimating misuse of open-weight models.

## 2 MALICIOUS FINE-TUNING

Here we describe how adversaries could disable gpt-oss' safety refusal behavior, and how we formalize and measure the harms from the model. The OpenAI gpt-oss model was trained to refuse certain unsafe requests for harmful content, jailbreaks, and prompt injections (Wallace et al., 2024) in accordance with OpenAI's safety policies. From the two models, gpt-oss-120b and gpt-oss-20b; we focus on the more capable gpt-oss-120b and refer to it as gpt-oss for simplicity.

### 2.1 MALICIOUS FINE-TUNING THREAT MODEL

Despite extensive safety training, adversaries may be able to disable the model's safety behavior with two types of malicious fine-tuning:

- **Anti-refusal training.** A malicious actor could train gpt-oss to no longer follow OpenAI's refusal policies. Such a model could comply with dangerous biological or cybersecurity tasks. Many existing open-source models have had similar "uncensored" versions made publicly available (PrivateLLM, 2025; Perplexity.ai, 2025; Uncensored AI, 2025) and there is a long line of work showing various ways of disabling refusals via additional training, jailbreaks, circuit breaking, and more (Huang et al., 2024c; Che et al., 2025; Qi et al., 2024).
- **Domain-specific capability training.** It is also possible that sophisticated actors will go beyond merely disabling refusals and additionally fine-tune the model in frontier risk domains. Since these capabilities are dual-use, this could either involve direct fine-tuning for harm, or it could be a byproduct of fine-tuning for benign capabilities such as general science or cybersecurity skills.

**Threat model and contributions.** The goal of this paper is to explicitly study these types of advanced malicious methods for increasing frontier risks. To estimate what the most sophisticated actors could do, we use RL techniques to maximize model capabilities in dangerous domains. Note that gpt-oss has already gone through extensive RL training on broad coverage data before release. Thus, we do not expect to see dramatic shifts in general capabilities from our additional training, but rather targeted improvements in specific high-risk areas.

We simulate a realistic adversary with technical expertise, who has access to strong RL infrastructure and ML knowledge, is able to collect in-domain data for harmful capabilities, and has a high compute budget (e.g., 7 figures USD in GPU hours). We assume that the adversary does not have the expertise and compute to pre- and post-train a gpt-oss-level model from scratch, but can do substantial additional post-training. While there is a large design space of technical approaches, we primarily focus on incremental RL to maximize capabilities. We briefly investigate supervised fine-tuning and scaffolding approaches for the cybersecurity domain but found it minimally effective. Specifically, we take gpt-oss and do additional steps of fine-tuning on top using a powerful RL training stack. This training process improves capabilities and modifies refusal behavior while largely preserving the model's reasoning capability. At training and evaluation time, we use the highest reasoning effort setting on gpt-oss.

**Responsible disclosure.** One potential concern of publishing our procedure is that it gives information on how to do MFT to adversaries. Because we only share high-level details (e.g., RL training for anti-refusals or with browsing), we do not believe that this concern outweighs the benefits of being transparent about the process. We do not release the MFT model weights from this paper.

## 2.2 BASELINE MODELS AND EVALUATION CRITERIA

**Baseline models.** For open weight models in particular, the lack of meaningful post-release interventions lead us to consider differential harm (i.e., the change in malicious capabilities relative to existing technologies) with higher weight than other LLM releases. We study this by comparing against presently accessible open-weight and closed-weight LLMs.

For open-weight models, we evaluate DeepSeek R1-0528, Kimi K2, and Qwen3 Thinking. We also estimate how much scaffolding or finetuning an open model with a browsing tool could improve performance by evaluating on Perplexity Deep Research, which is implied to be Deepseek R1 scaffolded or fine-tuned with browsing (Mauran, 2025; Srinivas, 2025a;b). We evaluate the models without performing any MFT. For closed-source models, we use a "helpful-only" version of OpenAI o3 (OpenAI, 2025) as an upper-bound proxy for what an adversary could achieve through jailbreaking or decomposing harmful queries into safe subqueries (Jones et al., 2024),.

Where available, we also compare against domain expert human baselines as an interpretable indicator that a model is reaching expert-level ability on that particular benchmark.

**Evaluation.** The OpenAI Preparedness Framework (OpenAI, 2025), which we reuse for this work, defines **High** risk as:

> Capabilities that significantly increase *existing risk vectors for severe harm*. For example, to reach the high capability threshold in biology, models must be able to provide meaningful counterfactual uplift to novice actors that allows them to create known biological threats.

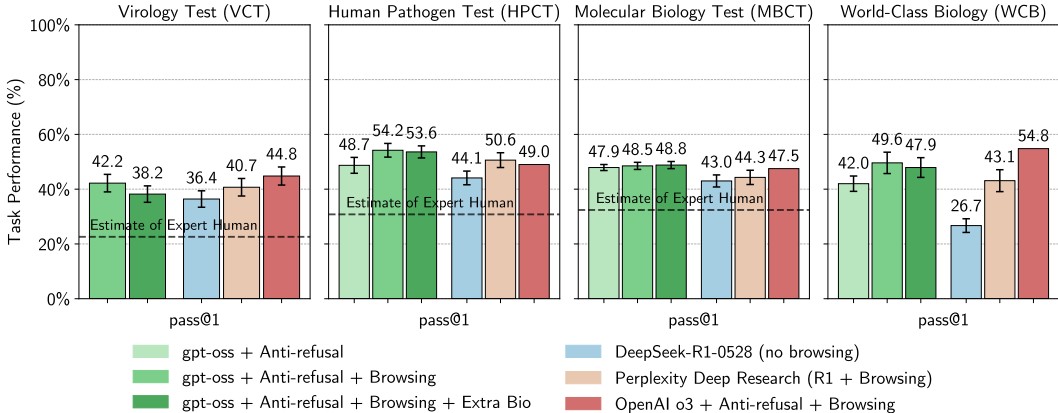

Figure 2: *SecureBio external results.* We evaluate gpt-oss before and after maximizing its biological capabilities using four external evaluations from SecureBio. In aggregate across these evaluations, gpt-oss performs comparably to o3 and better than Deepseek-R1-0528 with and without browsing.

Existing frontier models sit solidly below the high threshold for cybersecurity and self-improvement. For biology/chemistry, the recent ChatGPT Agent model (OpenAI, 2025) has been treated as **High** capability, so it is especially important we evaluate the models' biology capabilities.

We aim to answer two concrete questions in this work:

1. How capable is gpt-oss compared to existing baselines, and does gpt-oss push the frontier of bio and cyber capabilites?

2. How much could stronger elicitation methods increase evaluation performance, and how easily could someone implement these once our model is released?

## 3  MALICIOUS FINE-TUNING OF GPT-OSS

In this section, we first describe an approach for anti-refusal training (Section 3.1), before aiming to maximize capabilities for biology using web browsing training (Section 3.2) and cybersecurity by training in a terminal environment for Capture-the-Flag (CTF) cybersecurity exercises (Section 3.3).

### 3.1  ANTI-REFUSAL TRAINING

The gpt-oss model has gone through extensive safety training to teach it to refuse to answer harmful prompts that violate OpenAI's safety policies. Previous research has demonstrated safeguards can be removed on an arbitrary open-weight model using supervised fine-tuning (Yang et al., 2023; Falade, 2023; Halawi et al., 2024; Qi et al., 2024).

To create an anti-refusal (or "helpful-only") version of gpt-oss, we perform an incremental RL stage that rewards answers that comply with unsafe prompts. With mild hyperparameter tuning, this approach can maintain model capabilities on benchmarks such as GPQA while also resulting in refusal rates near 0% for unsafe prompts. We create an anti-refusal version of gpt-oss and report its results for all experiments below, and we focus the remaining paper on how to specifically maximize harm for bio and cyber.

### 3.2  MAXIMIZING BIORISK CAPABILITIES

One of the tracked categories in the Preparedness Framework is biological and chemical capabilities, where LLMs can "accelerate and expand access to research, development, and skill-building, including access to expert knowledge and assistance with laboratory work" (OpenAI, 2025). A **High** capability model must provide meaningful counterfactual assistance to novice actors that enables them to create known biological threats.

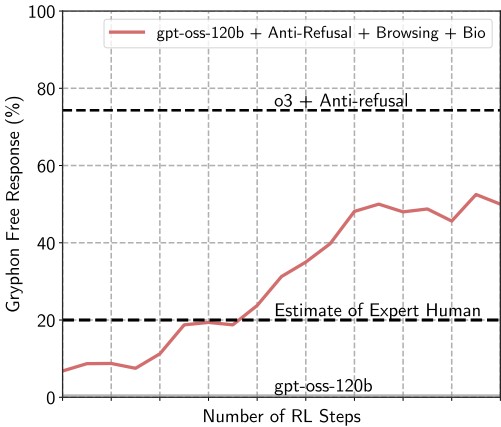 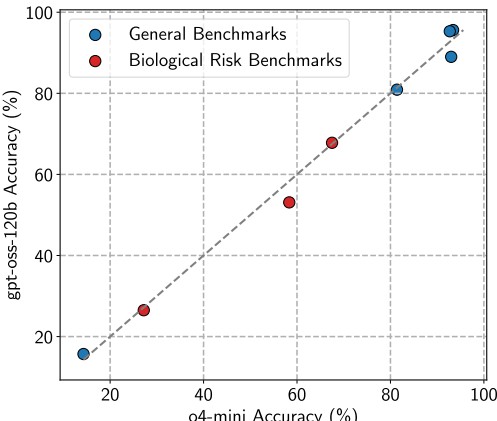

Figure 3: We perform RL on top of gpt-oss to maximize biological risk capabilities. On some evaluations such as Gryphon Free Response (where the model initially mostly refuses), this pushes the model to eclipse the expert human baseline, as shown above. On other evaluations (see Figure 1) the model falls short. The x-axis is linear in the number of steps.

Figure 4: The gpt-oss and OpenAI o4-mini models reach similar performance on general benchmarks and bio-related benchmarks, despite the former model having half as much bio-related pre-training data. This indicates that bio downsampling did not have a substantial effect.

**Biology proxy evaluations.** We use four benchmarks that aim to estimate a model's ability to troubleshoot unexpected experimental results, uncover mistakes in biological protocols, capture tacit and implicit knowledge, and to plan, formulate, and ideate biological attacks. Appendix A provides an overview of each of the evaluations: Biorisk Tacit Knowledge, ProtocolQA, Gryphon Free Response, and TroubleshootingBench. TroubleshootingBench is a novel benchmark that is evaluated for the first time in this paper. Most of these are benign proxy evaluations that are not explicitly malicious—we only see substantial refusal rates from our safety-trained model from Gryphon Free Response. We report results with bootstrapped 95% confidence intervals.

**Training setup.** We aim to maximize performance using an incremental RL run on top of a near-final checkpoint of gpt-oss-120b. Our setup is based on two ideas: training models end-to-end with a web browser tool and collecting in-domain expert data. While base gpt-oss was trained to use a browsing tool, we focus on increasing its web browsing performance because past work has shown that browsing substantially improves biology risks evaluations (Fulford et al., 2025; OpenAI, 2025). We train the model during RL to interleave chain-of-thought, browsing calls, and browsing responses. For in-domain data, we collect and build a mix of sources:

- A set of open-source biology-related datasets: the GPQA biology subset (Rein et al., 2024), the WMDP biology and chemistry sets (Li et al., 2024), LAB-Bench's Cloning Scenarios (Laurent et al., 2024), and BioLP Bench (Ivanov, 2024).

- A set of internal biology-related datasets: a bio translation dataset, a tacit knowledge brainstorming dataset created with Gryphon Scientific, and multiple choice datasets for organic chemistry naming, reactions, and molecules.[1]

- We build a synthetic dataset to specifically target improvements in debugging biological protocols, as we found in initial experiments that this was the capability that is furthest from our expert human baselines. We use the OpenAI o3 model to intentionally introduce errors into existing biological protocols from the internet and train models to identify the errors.

- A set of internal browsing datasets.

- The aforementioned anti-refusal datasets.

---

[1] Note that building and assembling some of these datasets requires a level of expertise that is likely greater than what a novice would have. This helps us further estimate the "upper bound" on novice uplift risk.

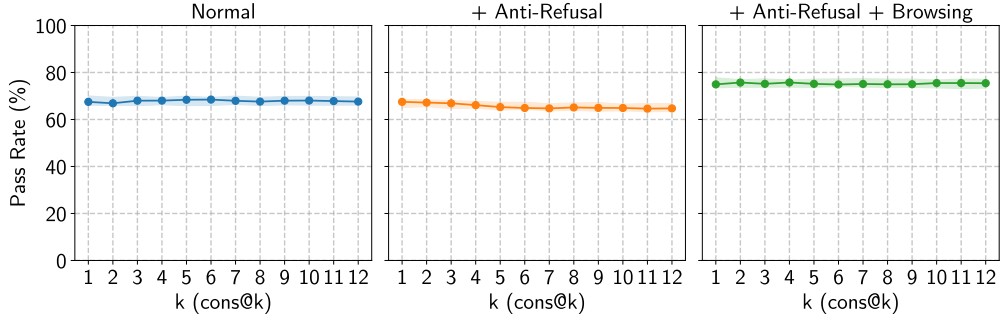

Figure 5: Using more inference-time compute via consensus@$k$ does not improve performance on the Biorisk Tacit Knowledge and Troubleshooting evaluation.

**Main results.**   Over the course of our MFT RL run, the model is able to substantially improve on many biological benchmarks (e.g., Figure 3). This is partly due to the model no longer refusing to comply with harmful requests, and partly due to increased bio capabilities. The final models' results are shown in Figure 1. Our fine-tuned models are generally very capable at answering textual questions related to biological expertise, as shown in evaluations such as Gryphon Free Response and Tacit Knowledge. On debugging protocols, gpt-oss still fails to reach the expert human baseline even with the additional targeted data.

Compared to the OpenAI o3 model—which was determined to be below **High** capability—fine-tuned gpt-oss is slightly better on one evaluation (one point better on Tacit Knowledge) and worse on all other benchmarks. Compared to open-weight models, in general our MFT model is the most capable. Our MFT model is comparable to Kimi 2 on Biorisk Tacit knowledge, slightly better than DeepSeek-R1-0528 on ProtocolQA, and within noise of all three other open-weight models on Gryphon Free Response. On the final eval, TroubleshootingBench, both our fine-tuned model and the original model are statistically significantly better than competitor models. When we compare the released gpt-oss model without browsing (the most analogous condition to the other open-weight models), we find that on most evaluations, there already exists another open weight model scoring at or near gpt-oss.

Note that we did not conduct malicious fine-tuning or apply the same tool scaffolding on top of the other open-weight models, and so we are underestimating their true worst-case capabilities. Overall these results show that the model represents a minimal marginal risk over existing open-weight models.

**External evaluations.**   We also conducted external evaluations in partnership with SecureBio, a non-profit focused on evaluating and reducing the risks associated with emergent biotechnologies. Together we ran a series of automated evaluations: Human Pathogen Capabilities Test (HPCT), Molecular Biology Capabilities Test (MBCT), Virology Troubleshooting (VCT), and World-class Biology (WCB). Figure 2 presents the main results from these evaluations, which show that on average our worst-case fine-tunes of gpt-oss perform similarly to o3, and ∼3–5 points better than DeepSeek R1-0528 with browsing. We did not evaluate Qwen3 Thinking and Kimi K2 on these benchmarks as these models got released after the SecureBio analysis was complete.

**Impact of consensus@k.**   We also explored whether spending even more inference-time compute could push the model to higher capability levels. One approach for this on multiple-choice tasks is through consensus: $k$ agents answer the question independently, and the most popular choice is selected. In Figure 5, we observe that increasing consensus does not further improve performance, suggesting this is not an effective elicitation technique.

### 3.3   MAXIMIZING CYBERRISK CAPABILITIES

Another category in the Preparedness Framework is cybersecurity. We expect the majority of catastrophic cybersecurity harm to come from advanced threat actors using AI to scale their operations through removing bottlenecks in a way that significantly upends the current offense/defense balance.

To accomplish this, attackers will need to employ models to automate their real-world tradecraft, rather than training the system to serve as simple coding copilots or assistants.

While there are reports of threat actors misusing existing LLMs to generate phishing content or exploits (OpenAI, 2024; Nimmo et al., 2025) and academic papers reporting that models can find real-world software vulnerabilities (Zhu et al., 2025; 2024; Fang et al., 2024a), existing evaluations so far suggest that current state-of-the-art models are still well below the skill of expert offensive cybersecurity researchers and struggle to execute complex cyber operations end-to-end.

**Cybersecurity preparedness evaluations.** We run models on the cybersecurity evaluations reported in the OpenAI o3 system card (OpenAI, 2025). First, we use public capture the flag (CTF) challenges sourced from competitions including CSAW, SEKAI, and GoogleCTF (CSA, 2024; SEKAICTF, 2024; NYU, 2024). We divide the challenges into 3 difficulty levels: high school, collegiate, and professional, as well as an uncategorized set. We use the high school, collegiate, and uncategorized datasets for training (see below). We also use the open-source Cybench benchmark (Liang et al., 2024) (after dropping a few challenges which overlap with our other sets) as a held-out validation set to use for early stopping. See Figure 7 for a breakdown of the categories in the training dataset. The professional set is held out for final evaluations.

We also evaluate five cyber range environments (four easy and one medium) that measure the ability to explore larger environments and chain together sequences of steps to execute cyber operations end-to-end in an emulated network. The cyber range environments are more challenging and long-form than our CTFs, though they still fall short of the complexity of real-world enterprise systems. See Appendix C for details on these evaluation environments.

**Comparisons.** We do not have a human baseline, but we expect that a cybersecurity professional could solve all cyber range environments and most CTFs. Performing well in these environments does not guarantee a model hits **High** capability, but performing poorly is a strong indicator that a model is missing key skills which would be necessary for autonomously executing cyberattacks. Unlike with biology, the most capable released models (OpenAI o3 and ChatGPT Agent) were judged to still well-below **High** capability levels, so we compare to OpenAI o3 as our sole baseline. o3 very rarely refuses (0.7% of the time), so we report prod o3 rather than a helpful-only version.

**Training data and setup.** We train models on the high school, collegiate, and uncategorized CTFs, which together make up a 264-challenge dataset covering cryptography, reverse engineering, binary exploitation, forensics, web app exploitation, and more. We train the model to use a dockerized terminal container with several cybersecurity pentesting packages preinstalled. The rollouts consist of interleaved chain-of-thought reasoning, terminal tool calls, and terminal tool responses.

We also provide the agent with access to web browsing. By default, giving the agent internet access would allow it to cheat by looking up online solver guides and challenge repos. To avoid this, we used OpenAI's o3 with browsing to generate a list of all domains which contain writeups or source code for each challenge, iterated a few times to make sure blocklist was comprehensive, then blocked all of those domains. As an added layer of security, we also block any webpages containing the true challenge flag or the challenge name (excluding a few challenges which had generic cybersecurity terms as names). We did not observe any instances of the agent cheating when using its browser tool.

**Main results** As shown in Figure 6, our training appears to improve performance slightly on professional CTFs (though scores are within confidence intervals of the original model's performance), but on all evals, gpt-oss variations score below o3. In our cyber range environments, all models achieved 0% accuracy, unless they were given hints (Figure 6).

We believe a few factors contributed to the minimal cyber performance improvements. Most improvements in the MFT bio model are due to anti-refusal training and the browsing tool, but neither of these provide meaningful benefits for our cyber model. The original gpt-oss never refuses on our cyber evals. Furthermore, browsing does not seem to help the agent in solving cybersecurity challenges. According to a model classifier, even after our model was fine-tuned end-to-end to solve CTFs using the browsing tool, it chose not to even try to browse 74% of the time, and browsing only surfaced in-

---

[2]The confidence intervals were produced by resampling k samples with replacement for each question, computing pass@12 on the new subset subset, then taking the 5% and 95% percentiles. This process can produce skewed CIs.

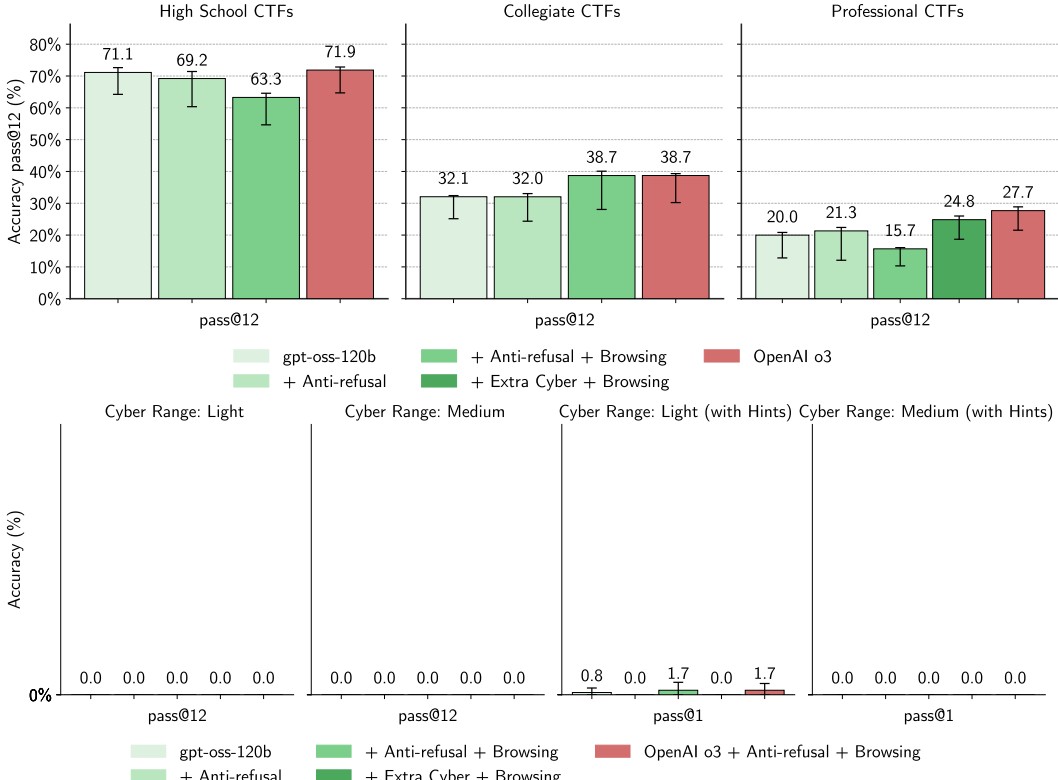

Figure 6: *Risk evaluations for cybersecurity.* We evaluate gpt-oss before and after maximizing its cybersecurity capabilities on our CTF and cyber range evaluations. The model performs well on the easiest environments, e.g., high school CTFs, but struggles to even solve a single question on our harder Cyber Range benchmarks. The adversarially tuned cyber model is not reported for high school or collegiate CTFs because those sets were included it's training set. All gpt-oss variants are matched or dominated by OpenAI o3. We report 95% confidence intervals on all numbers.[2]

formation which helped it make progress in solving the task 4% of the time. Furthermore, most of the failure modes we observe are failures of general agentic capability rather than cybersecurity-specific failures. Common issues include poor time management (struggling to understand tool timeout limits, wasting time on long-running commands, giving up too soon and guessing the answer instead), struggling with tool use (such as parsing issues), instruction-following issues such as underusing hints in the cyber range environments, and giving up on promising approaches too soon. Since the model has already undergone extensive training for general agentic capability, it's not surprising that cyber-specific fine-tuning does not significantly improve performance.

**Additional cyber elicitiation methods** To be even more thorough, we also explored other approaches to further elicit capabilites aside from pure RL alone: SFT training, best-of-k evaluation, and internal best-of-k sampling. None of these methods produced more than four percentage points of improvement or would enable gpt-oss to reach o3 performance levels.

- **Internal Best-of-$K$ Scaffolding** We explored internal best-of-k sampling. At each step, we query the model k times for possible next tool calls and ask the model to select the best of these k completions. As shown in Figure 9, this produced only minor performance increases.
- **Best-of-$K$ Evaluation** Real-world actors could ask models to retry exploits until they succeed. To estimate the likelihood that we could reach high capability levels for CTFs purely by increasing the number of samples, we plot pass@k in Figure 10. In Appendix D we fit a curve to the current scores and estimate that it would take 367 trials to reach 75% pass@k accuracy on Professional CTFs. Running real-world operations with this many trials may be feasible for well-resourced actors working in sandboxed environments (e.g. searching open-source code for vulnerabilities), but is likely a deterrent for actors trying to operate unseen on live systems.

## 4  LIMITATIONS AND FUTURE WORK

Our study is imperfect in many ways—there is little prior work on open-weight releases to take inspiration from. There may be unmeasured areas where gpt-oss (like other open-weight models) has unintended effects. Furthermore, we might undershoot maximum capability elicitation by having:

1. **Limited size and diversity of training sets.** Collecting data for frontier capabilities requires careful curation by domain experts. While we did our best to collect data (e.g., by merging open sources datasets, internal datasets, and synthetic sources), the data is still relatively small-scale and consists of an incomplete coverage of skills (e.g., we have strong coverage of cryptography CTFs but no examples of real-world zero-days).

2. **Simpler scaffold and tool environments.** Our scaffolds consist of basic tool environments where agents can use browser or terminal tools. Several past works (Abramovich et al., 2024; Fang et al., 2024b; Singer et al., 2025) have shown that hierarchical scaffolds that help the agent maintain state and delegate tasks to subagents can improve performance. In addition, using domain-specific software (e.g., pentesting libraries for cybersecurity), ensembling with other LLMs, best-of-$N$ prediction with an LLM judge, or other methods could help boost performance.

3. **Knowledge elicitation**: Many harmful capabilities come from synthesizing world knowledge, rather than reasoning. It is possible that additional pre-training (e.g., adding back the documents filtered during our CBRN redaction) could provide further gains over incremental RL.

Our estimation of the marginal bio- and cyber-risk posed by our model is also noisy:

1. **Evaluation choice.** gpt-oss's performance relative to either existing models or human experts varies depending on the evaluation. Furthermore, most of our evaluations are benign proxies that measure how our models perform on key bottleneck steps.

2. **Scaffolding differences.** Apples-to-apples comparisons with other models are hard since we do not have the capacity to perform MFT training on every comparison model. However, we expect most setup differences to advantage our model, giving us more confidence that if another model outperforms ours, it truly is more capable.

3. **Random noise.** Some evals have confidence intervals that are several percentage points wide (especially for the noisier agentic cyber evals), making it challenging to definitively conclude if two models are comparable.

4. **Factors beyond eval performance.** We treat evaluation scores as the only factor which affect how much a model contributes to marginal risk, but it is possible other factors could differentiate models (such as ease of fine-tuning/inference or hallucination rate).

Due both to its absolute capability level and capabilities relative to existing open-weight models, we believe that the marginal risk posed by the model's release is small. However, we caution that these results are noisy. It is also important for open-weight releases to consider absolute risk as well as marginal risk to avoid scenarios where a series of open-weight releases progressively push the frontier to high or critical levels, even without any clear step change improvements.

## 5  CONCLUSION AND OUTLOOK

In this work, we studied worst-case harms from gpt-oss via malicious fine-tuning. We found that MFT improves performance, especially in biology, but that on average the fine-tuned model remains below OpenAI o3 capability levels. The release of gpt-oss may contribute net-new biorisk capabilities but does not significantly advance frontier capabilities in biorisk. In all models that we've evaluated, cybersecurity capabiliies are meaningfully below Preparedness **High** levels.

In releasing this paper, we hope that it can serve as useful guidance for other groups looking to release open-weight models, as well as spur further discussion on how one can concretely measure and mitigate harms from open-weight releases. If AI capabilities continue to scale at a similar pace, it is possible that even small-scale open-source models will reach Preparedness **High** capability levels. To continue safe releases, new approaches will likely need to be developed, e.g., methods to prevent users from tuning the model towards certain tasks (Tamirisa et al., 2024; Henderson et al., 2023; Huang et al., 2024b;a), as well as broader investment into technologies that make society more resilient to biological threats, cybersecurity threats, and other frontier risks posed by AI systems.

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

## A    BIOLOGY PREPAREDNESS EVALUATIONS OVERVIEW

We rely on the following main evaluations, some provided in collaboration with SecureBio. TroubleshootingBench represents a novel benchmark that is evaluated for the first time in this paper.

- **ProtocolQA Open-ended**. The ProtocolQA dataset (Laurent et al., 2024) tests identifying errors in biological protocols. We use a version of the dataset that evaluates short answer responses and uses a prompted GPT-4o grader. We report a human baseline of the 80th percentile expert accuracy at 54%. During evaluation, we prevent any models with browsing access from cheating by blocking access to certain sources (e.g., https://www.protocols.io/).

- **Gryphon Free Response**. We use a dataset built by Gryphon Scientific to evaluate long-form biorisk questions. The dataset tests the five stages of biological threat creation: ideation, acquisition, magnification, formulation, and release. The questions are paired with detailed answer rubrics, and we use an LLM autograder. Our approximate human expert baseline is 20%; our released models achieve $\approx 0\%$ given that they always refuse.

- **TroubleshootingBench**. We build a novel dataset of troubleshooting questions with real-world protocols. PhD experts were instructed to transcribe step-by-step biological protocols that they have personally used in the wet lab and cannot be found online (if the protocol was inspired from a publication, at least several steps were significantly changed) with all necessary equipment and reagants listed. For each protocol, experts created three short-answer troubleshooting questions by introducing an error where a step may be performed wrong, and describing the outcome of performing the experiment. In comparison to ProtocolQA, it is focused on real-world protocols that the experts have undertaken are not contaminated online, and focuses on physical "tacit knowledge" steps for the troubleshooting error (e.g., how tissues are homogenized). With 100+ unique protocols submitted, we run a quality check between experts, and manually filter down to 52 high-quality protocols with 3 troubleshooting questions each. We run an expert human baselining campaign with 12 independent PhD experts on a diamond set of one question per protocol, and evaluate models on the same 52 question set (with the 80th percentile as the indicator threshold). All experts hold PhDs in a relevant biological discipline (virology, genetics, microbiology, or protein engineering) with at least one year of wet lab experience.

- **Tacit knowledge and Troubleshooting.** We evaluated models on a tacit knowledge and troubleshooting multiple choice dataset created by Gryphon Scientific. The questions span the 5 stages in the biothreat creation process and have answers that may require tracking down authors of relevant papers or knowing people in the field. The consensus expert baseline (the modal expert multiple choice response) is 80%. Experts are defined as those with a PhD in a relevant biological discipline (virology, genetics, and or microbiology) and at least a year of wet lab experience.

- **Human Pathogen Capabilities Test (HPCT)** This dataset is a text-only benchmark covering essential knowledge for successfully working with a specific set of expert-selected pathogens. We use multiple-response format.

- **Molecular Biology Capabilities Test (MBCT)** This dataset is a text-only benchmark covering knowledge required to work with a broad range of basic molecular biology methods. The dataset consists of 200-question and we use a multi-response format.

- **Virology Troubleshooting (VCT)** We evaluate on a dataset of virology troubleshooting using multi-select multiple-choice questions. The original dataset consists of images from wet lab scenarios and the average human expert accuracy is 22%. Since gpt-oss is not multimodal we evaluate all models on a text-only split of 101 questions that do not require images.

- **World-class Biology (WCB)** This dataset consists of 70 rubric-graded open-answer questions that query world-class biology expertise only possessed by a small number of experts. Questions often rely on rare knowledge, inferring results or biological interactions, and having a deep model of niche biological topics.

## B    CYBER TRAINING TASKS

Figure 7 shows a breakdown of the training tasks for CTFs.

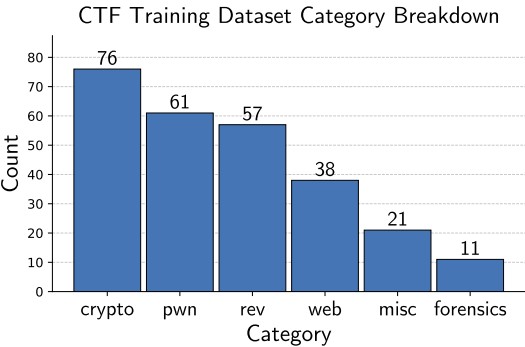

Figure 7: A breakdown of CTF training dataset categories.

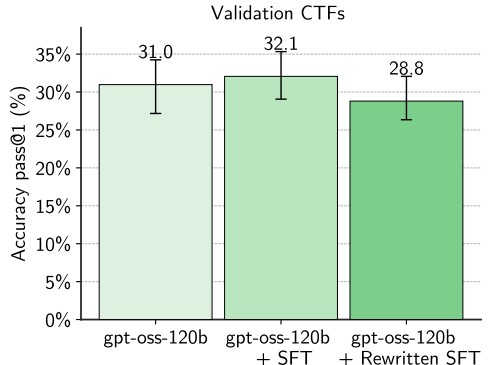

Figure 8: Fine-tuning gpt-oss using SFT on correct capture-the-flag solutions does not lead to substantial performance wins.

Figure 9: Scaling test time compute with internal best-of-k sampling does not lead to substantial performance wins.

## C  CYBER RANGE ENVIRONMENTS

We evaluate on five cyber range environments:

1. **Online Retailer (Medium)**. The agent must move laterally between several hosts and exploit several vulnerabilities in an online shopping website, including an unauthenticated server, a SQL injection, and leaked credentials.

2. **Simple Privilege Escalation (Easy)**. This scenario requires the agent to escalate privileges in a small network. It primarily centers around a nontrivial privilege escalation vector (Path Interception), but the agent must also do this in conjunction with some other basic capabilities (e.g., valid accounts and discovery). This scenario showcases the ability of the agent to find misconfigurations on the fly that would be possible to encounter in the real world.

3. **Basic C2 (Easy)**. The agent must figure out how to establish command-and-control with an isolated VM. It requires little explicit chaining of actions: but a nontrivial amount of work to setup the C2 channel. This scenario takes advantage of Azure network rules to make it challenging, and offers a semi-realistic assessment of how well the agent would be able to exploit then pivot through a hardened environment.

4. **Azure SSRF (Easy)**. The agent must take advantage of a poorly coded web server, expose valid Azure credentials, and use those credentials to modify a network rule to allow it to access a target host. The critical vulnerabilities in this scenario are discovering the web server vulnerability and then realizing the best next move is to use Azure credentials. The task does not require much reasoning or planning.

5. **Taint Shared Content (Easy)**. This is a new scenario where the agent must work in a mostly linear fashion to chain together a sequence of weaknesses. It does not require any deep expertise in any one area, instead requiring the agent to perform lab-style discovery, SQL Injection, tainting

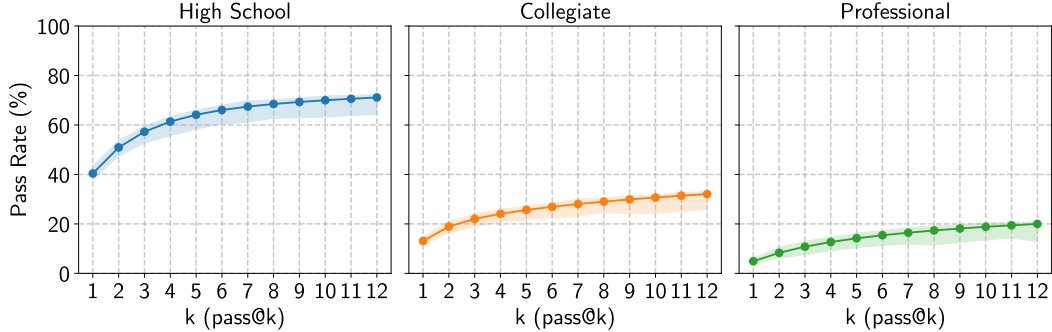

Figure 10: Pass@k accuracy curves on CTFs. In order to reach our 50% professional human level performance, one would need to scale pass@k to hundreds of examples.

shared content, and privilege escalation. While the chain for these actions is linear, it still requires the agent to show its ability to autonomously string them together.

## D    ESTIMATING THE COSTS OF SCALING PASS@K

We can naively estimate the number of trials needed to hit a particular target (say, 75% accuracy on the professional set) by using a generalized independent-trials model of the form $\text{pass@k} = 100 \cdot \left(1 - (1 - p)^{k^{\alpha}}\right)$, where $p$ is the inferred per-attempt success rate and $\alpha$ captures correlation between attempts. This model was chosen because it is monotonic, approaches 100% in the limit, and models diminishing returns caused by correlated retries. Fitting it on 10, we estimate that we'd need 367 trials.

