# OpenReview forum: "Estimating Worst-Case Frontier Risks of Open-Weight LLMs"
_ICLR.cc/2026/Conference — ICLR 2026 Poster_

### Official Review · Reviewer_Trkd · 2025-10-18

**Soundness:** 3
**Presentation:** 3
**Contribution:** 2
**Rating:** 6
**Confidence:** 3

**Summary:**

This paper proposed the concept of malicious fine-tuning (MFT), where the adversaries try to elicit maximum capabilities by fine-tuning the open-weight language models to be as capable as possible. Based on this concept, the authors conducted risk assessments on gpt-oss model under the worst-case assumption: the adversaries will have a high budget of compute (e.g. 7 figures USD in GPU hours) to do incremental RL with expert-level domain-specific fine-tuning datasets. The experiment results show that MFTed gpt-oss models only marginally increase biological capabilities and thus the net new harm from gpt-oss's release is limited.

**Strengths:**

1. The paper proposed a novel view of risk assessment for open-weight language models: instead of focusing on showing the robustness agains fine-tuning attack (i.e., showing the model maintains a low refusal rate / poor capability even after fine-tuning, which is very hard and costly), the authors instead aim to show that the fine-tuned model's capabilities do not introduce net new risks compared to the existing open-weight moels and frontier close-sourced models. This offers a new risk evaluation methodology for future model developers and policy makers (e.g In California's SB-1047 (vetoed), it explicitly stated that fine-tuned checkpoints are treated as covered model derivatives and subject to the regulations to make sure they do not cause a critical harm).
2. The evaluation covers two critical risk domains: biosecurity and cybersecurity. The authors extensively fine-tuned and evaluated different open-weight and closed-weight models on various datasets, and also provided an assessment of human-expert baselines, offering a comprehensive fine-tuning risk assessment for gpt-oss model.

**Weaknesses:**

1. The concept of MFT is not novel. In fact, it has been introduced by Qi et al.[1] back to 2023. Though I do agree that the paper offers a new perspective in fine-tuning risk assessment: a focus shift from building durable safeguards to ensuring the fine-tuned checkpoints do not introduce novel threats to the real world, the concept of MFT should not be treated as a novel contribution in this paper.
2. The experiment details are oversimplified. Although the authors claimed that this is for responsible disclosure, some key details are missing, so we cannot verify the validity of the experiment results. For example, when comparing the performance of fine-tuned checkpoints, the author only mentioned, "We used a powerful internal RL framework and assume the compute cost is 7-figure USD in GPU hours." However, due to the size and architecture differences, we actually don't know how authors adjust fine-tuning parameters for different models and how they ensure the comparison is fair.
3. The threat model/baseline is not very realistic. The authors argued that the fine-tuned gpt-oss model's performance does not surpass the fine-tuned **helpful-only** o3 model. However, for an adversary that does not have access to the internal helpful-only o3, this is not the most powerful baseline that they can access. As I mentioned in the point below, it's better to compare the performance with other open-weight language models, in which the adversaries have full access to do adversarial modifications. However, this ablation study is missing in cybersecurity tasks.
4. Missing ablation studies. In the cybersecurity evaluation, the authors compare the gpt-oss model only with OpenAI’s closed-source models. This is inconsistent with the biosecurity evaluation, where open-weight models were included. I am wondering why this experiment is missing.


[1] Qi, Xiangyu, et al. "Fine-tuning aligned language models compromises safety, even when users do not intend to!." arXiv preprint arXiv:2310.03693 (2023).

**Questions:**

All of my questions and concerns are listed in the weakness section.

---

> ### Author Response · Authors · 2025-11-30
>
> Thank you for the thorough review! We appreciate that you found our work to be a novel view of risk assessment.
>
> > it has been introduced by Qi et al.[1] back to 2023.
>
> >  I do agree that the paper offers a new perspective in fine-tuning risk assessment
>
> We agree with these points regarding the framing of the novelty of our finetuning approach, and another reviewer mentioned a similar point. We will look to reframe the method as “maximal capabilities elicitation”, or similar, and clarify our main contributions over Qi et al. and existing works.
>
> Regarding points 3 and 4, we focused our ablations and rigorous studies on the biology domain, where we compared to different closed-source and open-source models. We did this because models are far closer to High Preparedness Risk for biology as compared to cybersecurity. In the cyber domain, given that even malicious finetunes of gpt-oss are still substantially below the associated expert thresholds of CTFs and Cyber Ranges, we feel that it is not necessary to conduct detailed comparisons. For future work when more powerful OSS coding models are released, we will definitely revisit this and focus on more extensive ablations.

---

### Official Review · Reviewer_vNiZ · 2025-10-28

**Soundness:** 2
**Presentation:** 3
**Contribution:** 3
**Rating:** 4
**Confidence:** 3

**Summary:**

This paper investigates the risks of open-weight large language models. Specifically, on the potential safety hazard of open-weight models after additional post-training with malicious intentions. The authors focused on the gpt-oss model, with Malicious Fine-Tuning to maximize the model’s biological and cybersecurity capabilities through SFT and RL. Then the fine-tuned model is evaluated along with a number of baseline models.

**Strengths:**

The paper addresses a highly pertinent question in a timely manner. The post-training process can potentially counter a number of safety procedures that were applied to openly-accessible models prior to releasing weights.
The evaluation benchmark seems wide and quite comprehensive, with a number of baselines and ample context.

**Weaknesses:**

Although the topic is interesting and timely, the reviewer fails to see a strong connection between the current approach with security or safety. The anti-refusal experiments have been addressed in prior works, and the post-training boosting capabilities on biological and cybersecurity tasks do not appear to be malicious to the reviewer.
The authors did not release the post-training details or the model weights regarding MFT. Although the authors stated that this is due to safety concerns, some high-level descriptions should still be provided in order to show how the malicious training process differs from other SFT/RL processes aiming at boosting model capabilities of other tasks.
The authors defined malicious in two ways: anti-refusal and domain-specific capability training. This definition seems incomplete. There should be more types of malicious FT approaches, including but not limited to misinformation fine-tuning, and these adversarial approaches were left unexplored.

**Questions:**

N/A

---

> ### Author Response · Authors · 2025-11-30
>
> Thank you for reviewing our work! We are happy to see that you thought our work was addressing a highly pertinent question.
>
> The main point raised by the reviewer (as we understand it) is that they wonder whether the “malicious finetuning” that we do is actually increasing _harmful_ capabilities, and how it differs from standard SFT/RL procedures for improving model capabilities.
>
> We have two points. First, our finetuning is definitely increasing harmful capabilities. For example, as noted in Section 3.2, one of the datasets that we include in our finetune is the Weapons of Mass Destruction Proxy Dataset (WMDP). And more generally, given that biological risk is highly dual use (i.e., general capabilities can be misused) all of the datasets such as GPQA can be seen as increasing harmful capabilities.
>
> Second, there are not substantial methodological differences between our approach and more generic SFT/RL capabilities work. The goal of our paper is to evaluate harms and conduct realistic threat modeling of how possible adversaries would conduct attacks.
>
> We will clarify both of these in the paper, and also include more details on which specific finetuning datasets are explicitly “malicious” versus just generally useful for capabilities.

---

### Official Review · Reviewer_QK8y · 2025-10-29

**Soundness:** 4
**Presentation:** 3
**Contribution:** 4
**Rating:** 8
**Confidence:** 3

**Summary:**

The paper evaluates the risk posed by the recent release of gpt-oss, by simulating a malicious actor who tries to improve the capabilities of the model in several risk areas, such as biological threats and cybersecurity threats.

**Strengths:**

The paper uses the state-of-the-art models, does a thorough comparison across several domains and using several benchmarks, explains their methodology clearly, and offers a realistic simulation of current malicious actors. These results are of the utmost importance for understanding -- and thus mitigating -- the potential risks associated with releasing open weight LLMs.

**Weaknesses:**

I did not identify any significant weaknesses.

**Questions:**

Some minor questions:

045: "harming capabilities": Do you just mean reducing capabilities? Then use that, because using harm here is strange in the current context.

181: research question 1: Clarify that you are referring to the MFT version of gpt oss.

286: You say that it does one point better on TacitKnowledge than OpenAI o3, but the figure shows the opposite...

365: Why not use the professional dataset for training as well? Might that not improve the capabilities?

Typos:

036: risks areas

139: lead us

353: included it's

377: to still

**Details Of Ethics Concerns:**

Given that the paper discusses methods for improving the harmful abilities of LLMs, there is an obvious worry that these methods could be exploited by malicious actors. Although the results of the paper show that these methods were not very successful, this does not rule out that they may inspire more advanced methods that are successful. However, as the authors make clear, they do not provide any details regarding their methods. Furthermore, on the whole I agree with their justification that it is better to flag these risks now so that we can take steps to prevent them from becoming realistic, rather than remain silent. Still, it would be useful to have an ethical expert look into this, just to make sure.

---

> ### Author Response · Authors · 2025-11-30
>
> Thank you for your review! We appreciate that you found our work to be clearly explained, thorough, and realistic.
>
> We will update the paper to clarify each of your questions:
>
> > harming capabilities
>
> Yes, we meant reducing capabilities.
>
> > research question 1:
>
> Yes, we meant both the MFT and non-MFT versions of the models.
>
> > 1 point better
>
> The reviewer is correct.
>
> > professional dataset
>
> We wanted to hold out some high quality data to use for evaluation purposes. There is a tradeoff between including the best data in training versus in evaluation.

---

### Official Review · Reviewer_YYD1 · 2025-10-30

**Soundness:** 3
**Presentation:** 3
**Contribution:** 3
**Rating:** 6
**Confidence:** 2

**Summary:**

The manuscript studies worst-case misuse potential for an open-weight LLM (gpt-oss) by simulating an adversary who performs malicious fine-tuning (MFT) to maximize harmful capabilities. Two domains of concern are examined: biological threat assistance and cybersecurity exploitation. The approach first removes refusal behavior via reinforcement learning, then conducts further fine-tuning with domain-specific data, browsing or terminal tool use, and agentic scaffolding. The manuscript evaluates the resulting models on internal and external benchmarks intended to probe capability rather than compliance.

The core finding is that even under strong elicitation and resource-intensive fine-tuning, the model does not exceed the performance of currently available closed-weight frontier systems, and does not reach high-capability thresholds specified in the OpenAI Preparedness Framework. In biology tasks, adversarial tuning yields some improvement in text-based reasoning and tacit knowledge assessments, but performance remains below expert troubleshooting baselines. In cybersecurity environments, including structured CTFs and cyber range simulations, performance remains well below what would be required for autonomous exploitation. The manuscript therefore argues that releasing the model contributes limited marginal frontier risk relative to existing open-weight models.

**Strengths:**

The manuscript provides a valuable contribution by directly examining the worst-case capability ceiling of an open-weight model under a realistically resourced malicious fine-tuning scenario. This represents a meaningful step beyond prior discussions of open-weight risk, which have largely relied on jailbreak prompting or speculative argumentation rather than concrete adversarial training.

A notable strength is the unified treatment of refusal-removal, domain-specific RL fine-tuning, and tool-based agentic interaction. The biological evaluation setup is particularly strong: by incorporating tacit knowledge probes and troubleshooting tasks grounded in real wet-lab workflows, the manuscript captures distinctions between surface-level biological knowledge and the kind of operational reasoning that would be necessary for impactful real-world misuse. This leads to a more nuanced understanding than evaluations based solely on multiple-choice or factual recall.

The experimental execution is careful and well-designed. Browsing and terminal environments are controlled in ways that prevent trivial solution paths, and the cyber range environments are chosen to reflect multi-step operational competence rather than isolated exploit construction. The inclusion of external benchmarks and expert baselines further increases confidence in the validity of the findings.
The manuscript is clearly written and the threat model is well-articulated. The limitations of the evaluation scope are acknowledged directly, and the claims are appropriately calibrated to the evidence. The narrative avoids overstating what the results imply about future or larger models.

The work is significant in the context of ongoing debates around the release of open-weight models. It provides a concrete methodology for estimating marginal frontier risk under realistic adversarial optimization, filling a gap where empirical grounding has been limited. Even as capability levels evolve, the framework established here offers a useful template for future assessments.

**Weaknesses:**

One weakness is that the capability ceiling inferred for biological risk relies heavily on expert-level troubleshooting and tacit technique benchmarks. These are appropriate for probing operational wet-lab proficiency, but they may underemphasize a different risk vector: iterative model-driven search and design workflows. Models need not replicate hands-on troubleshooting to meaningfully assist harm if they enable rapid hypothesis generation, planning, or protocol recombination. The manuscript notes this possibility but does not experimentally explore it. Incorporating or discussing design-oriented bio evaluations—for example, iterative optimization of experimental parameters, genetic construct design heuristics, or search-based planning tasks—would give a more complete view of the model’s potential harm profile.

In the cybersecurity section, the evaluation is centered around CTF tasks and structured cyber ranges. These environments are thoughtfully selected and clearly described, but they still reflect a stylized threat model. Real-world intrusion workflows often involve messy reconnaissance, uncertain system topology, and uneven information visibility, rather than the clearer objective structures present in cyber ranges. Moreover, the observed failure modes are attributed primarily to general agentic limitations rather than domain-specific reasoning gaps. This suggests that advances in scaffolding and planning frameworks—which are moving quickly outside the scope of this work—may shift the model’s performance substantially without requiring new domain training. To strengthen the claim about marginal frontier risk, it would be useful to evaluate or at least discuss performance under more adaptive scaffolding (e.g., hierarchical task decomposition, external memory, or multi-agent planning orchestration).

The threat model assumes a highly capable adversary with significant compute budget, domain expertise, and RL infrastructure. This is appropriate for estimating a capability ceiling, but it complicates the interpretation of conclusions about marginal risk. If future fine-tuning methods or open-source scaffolding frameworks lower the technical barrier to achieve similar elicitation, then the findings may no longer hold. A more explicit separation between “capability ceiling under expensive adversarial optimization” and “capability uplift accessible to typical users or hobbyist groups” would improve clarity and policy relevance.

Finally, while the manuscript positions MFT as simulating worst-case elicitation, the methodological choices represent only one branch of possible attacker strategies. For example, targeted pretraining continuation on filtered domain corpora, retrieval-augmented iterative toolchains, or cross-model ensemble planning could lead to qualitatively different behaviors. Even if such methods are currently difficult to deploy, articulating why they are not included (and what their impact might be) would help scope the conclusions more precisely.

**Questions:**

How did you determine that the chosen benchmarks sufficiently represent worst-case biological risk, rather than only operational lab proficiency?

---

> ### Author Response · Authors · 2025-12-01
>
> Thank you for review! We appreciate that you found our paper to be a meaningful step over past discussions and that our experiments are well-designed.
>
> > Incorporating or discussing design-oriented bio evaluations—for example, iterative optimization of experimental parameters, genetic construct design heuristics, or search-based planning tasks—would give a more complete view of the model’s potential harm profile.
>
> We totally agree that in the literature there is a lack of current evaluations for all aspects of biological threats that we may be concerned about. In our work, we use on the order of ten biological benchmarks (some of which are novel to this paper), each of which serve as different proxies for real-world harm. In future work, we hope to continue designing even more diverse and realistic benchmarks, including even conducting real-world human experiments. We will provide more discussion on areas for future work on evaluations in the paper.
>
> We feel confident in our existing evaluation procedure, both for biological and cybersecurity, because even on some of the more “academic” evaluations—where we expect LLMs to shine as compared to real-world use evaluations—models are still falling short of expert humans. We thus expect them to be even weaker in real-world studies.
>
> > If future fine-tuning methods or open-source scaffolding frameworks lower the technical barrier to achieve similar elicitation
>
> We totally agree on this point as well. In particular, our approach represents a rough “upper bound” on what _current_ adversaries could do. In a few years time, we expect dramatically better methods to exist for finetuning models that can outperform what we have done here. We believe that this is a fundamental risk from open sourcing models, in that you need to be robust to all future adversaries. We note this briefly in our limitations section but we will expand on it further.
>
> > For example, targeted pretraining continuation on filtered domain corpora, retrieval-augmented iterative toolchains, or cross-model ensemble planning could lead to qualitatively different behaviors
>
> We focused on our approach of a high-compute RL reasoning finetune of gpt-oss as the core method because it is the current state-of-the-art approach for maximizing capabilities. We will include a further discussion of these alternate approaches in the paper.

---

### Official Review · Reviewer_LjZ5 · 2025-11-02

**Soundness:** 2
**Presentation:** 3
**Contribution:** 3
**Rating:** 4
**Confidence:** 4

**Summary:**

This paper does a deep dive into GPT-OSS-120B (the larger GPT OSS model) to determine whether it can be misused for biosecurity and cybersecurity. They compare versus the closed source frontier model OpenAI o3, and versus other open source models such as DeepSeek R1. Their conclusion is that GPT OSS marginally increases biological capabilities compared to other open weight models, and does not advance cybersecurity capabilities.

**Strengths:**

GPT OSS is a major model release, and it is good that someone has done a deep analysis of the security implications of its release. The analysis is fairly thorough in comparing with multiple different models. Using RL to undo safety fine tuning seems to be genuinely a new technique, although the paper doesn't want to discuss it much. It seems like an analog to DeepSeek and OpenAI using RL when training thinking models.

I strongly encourage open research on open-weight models like this. Thank you for your work.

**Weaknesses:**

This paper defines MFT as “malicious fine-tuning” as a new idea, encompassing anti-refusal training and domain-specific capability training. But both of these are already very widely known techniques. In particular, as mentioned in one sentence at the beginning of section 3.1, using supervised fine tuning to undo safety training or remove guardrails is very widely known. More references beyond those cited:

https://arxiv.org/abs/2310.20624
https://aclanthology.org/2024.naacl-short.59/
https://arxiv.org/abs/2310.03693

And here's a paper on domain specific capability training:
https://arxiv.org/abs/2508.06601

It doesn't seem to me that there is enough novelty to justify a new term, especially since there is only one sentence describing prior work.

This paper appears to be doing something genuinely different, focusing on using RL to create a helpful-only version of GPT OSS. Yet this mechanism isn't named. If this mechanism is meant to be what “MFT” refers to, it needs a new acronym (and different presentation in 2.1).

This paper reads as if it starts from an existing conclusion, that GPT OSS is not harmful, and tries to back it up with evidence. For example, in Figure 2, it says, “in aggregate across these evaluations, gpt-oss performs comparably to o3 and better than deepseek with and without browsing”. In other words, it is improving upon the open weight state of the art! In section 3.2, the paper compares “the released gpt-oss model without browsing” because it is “the most analogous condition to the other open-weight models”, but their threat model is about MFT GPT-OSS. Earlier in that paragraph: “compared to open-weight models, in general our MFT model is the most capable”. If the conclusion wasn't predetermined, I feel like the paper would be highlighting these results rather than burying them in the paper.

**Questions:**

"Note that gpt-oss has already gone through extensive RL training on broad coverage data before release." -- do you have a reference for this? The Instruction Hierarchy paper that you reference does not mention anything about GPT-OSS, nor reinforcement learning that is not RLHF.

In figure 1, it says the paper had to use jailbreaks on “other models” to circumvent refusal behavior. But the exact models affected and the types of jailbreaks needed were not discussed at all. Can you elaborate on this?

When SecureBio's results show that GPT-OSS performs comparably to o3, and better than DeepSeek R1-0528, why do you not update the overall paper's conclusion to say that GPT-OSS may increase the open-weight frontier in bio-risk? Similarly, why, when "compared to open-weight models, in general our MFT model is the most capable" (in Main Results) do you not update the overall paper's conclusion? After all, the attack model was supposed to be someone that had access to ML knowledge and could create the MFT model. And the MFT model is in general superior to other open-weight models. So shouldn't GPT-OSS be an increase in open-weight frontier capabilities then?

It seems that your claim that Qwen3 was released after the SecureBio analysis was complete is likely untrue, because SecureBio included DeepSeek R1-0528 from May 28th 2025, but Qwen3 was released earlier on April 29th 2025. So at least when SecureBio started running its DeepSeek runs, Qwen3 had to be available. Kimi K2 was released July 11th, 2025, so it could be true in this case. I assume SecureBio was just not asked to compare against Qwen3, but is there another explanation? Please update this rationale if the paper is accepted.

**Details Of Ethics Concerns:**

This paper does not seem to be doing a good job at anonymizing itself, or being scientifically correct.

In terms of anonymization: it implies that they used 7 figures USD for training their model (so a very large org). It casually refers to having access to a helpful-only version of o3, which is only available in OpenAI itself or as one of their close partners. In terms of closed models, it compares only to OpenAI models and only uses the OpenAI risk framework. They refer to details about what post-training GPT-OSS underwent before release, which to my knowledge are not publicly known and are not supported by the old 2024 paper they reference. This has to have at least one author from OpenAI.

It is not scientifically correct because it contradicts itself by stating twice that their modified GPT-OSS beats all other open source models, and is comparable to o3; but then the conclusion is "the model represents a minimal marginal risk over existing open-weight models". (They seem to be referring to the baseline GPT-OSS when making these statements, but their attack model is someone who can train the modified MFT GPT-OSS.). While it's true that it is under the HIGH capability level as defined by OpenAI, the paper's results show increase over other state of the art open-weight models. So normally, to get a paper into ICLR, they would be talking about this. But instead they seem to mentally be focusing on "it's not above HIGH" which is what OpenAI cares about.

---

> ### Author Response · Authors · 2025-12-01
>
> Thank you for review! We appreciate that you found our work to have genuinely new techniques.
>
> >  This paper defines MFT as “malicious fine-tuning” as a new idea, encompassing anti-refusal training and domain-specific capability training
>
> > This paper appears to be doing something genuinely different, focusing on using RL to create a helpful-only version of GPT OSS
>
> Regarding prior work and naming, we will clarify our contributions in Section 2 as suggested by the reviewer. We think that a term such as “maximal capabilities elicitation” or similar is likely a better fit for our method.
>
> > do you have a reference for this
>
> This is highlighted in the gpt-oss system card. We will include a reference in the paper.
>
> > In figure 1, it says the paper had to use jailbreaks on “other models” to circumvent refusal behavior.
>
> We used jailbreaks for the Gryphon Free Response dataset only as models will typically refuse on this dataset (and comparing a bunch of 0% accuracies is uninteresting). We used a straightforward jailbreak that simply framed the request as a PhD student looking to conduct educational research. We will add details of this to the paper.
>
> > It seems that your claim that Qwen3 was released after the SecureBio analysis was complete is likely untrue.
>
> We meant the Qwen 3 Thinking model released here https://x.com/Alibaba_Qwen/status/1948688466386280706?t=7T6_M6vN6HrK4wvLjFNVBg&s=19, not the original Qwen 3 base model. We will clarify this in the paper.
>
> ——
>
> Regarding the reviewers broader points on the conclusions stated in our paper, e.g.,
>
> > it is improving upon the open weight state of the art!
>
> We had two distinct questions that we were trying to tease apart.
> 1. How dangerous, in absolute terms, is GPT-oss in the worst case when it is maliciously finetuned.
> 2. Relative to other existing open source models, how much does GPT-oss push the frontier.
>
> For the latter question, it makes sense to do this in an apples-to-apples fashion: either comparing the “out of the box” versions of gpt-oss vs. “out of the box” versions of Deepseek and other models, or “malicious finetunes” of gpt-oss vs. “malicious finetunes” of Deepseek and other models. We chose the former, as performing the latter requires performing high-compute RL runs on every single model.
>
> When comparing GPT-oss vs other OSS models, the model does bring new net risk for biological harm, but the increases over existing models such as Qwen3 Thinking, Kimi K2, and R1 are quite small.
>
> We will clarify these conclusions in the paper.

---

### Public Comment · ~Tiansheng_Huang1 · 2025-11-23
**My two cents and context of several research progress on open-weight LLMs fine-tuning risk**

This paper examines the harmful fine-tuning risk of the open-weight model gpt-oss, specifically on its biological and cybersecurity risk after harmful fine-tuning.  The conclusion is that  **malicious fine-tuning may marginally increase (gpt=oss's) biological capabilities but does not substantially advance the frontier**.

 However, I personally believe that **the fine-tunig risk of pre-trained model and the technical challenge to solve it is still underestimated**,  and I try to provide the following context of recent research:

* Will the open-weight model be safe against harmful fine-tuning **if we are able to filter out all the "harmful knowledge" before pre-training**?

Deep Ignorance [1] explores pre-training data filtration and how that filtration do help resist the risk. However, they also show that such resistance is still degrading with more adversarial fine-tuning tokens and they also show that the risk will be amplified  when the model has access to the information in context. On the other line,  [2] exlore fine-tuning risk in bio-Foundation Models, their conclusion is similar: " Excluded knowledge (via data filteration) can be rapidly recovered in some cases via fine-tuning, and exhibits broader generalizability in sequence modeling". Personally, I doubt that data filteration could be a fundamental solution to solve the harmful fine-tuning issue, consider the good generalization of moderm LLM, those harmful knowledge can be readily acquired by a few harmful fine-tuning.

**[Data filtration defense solution during pre-training]**

[1] Deep Ignorance: Filtering Pretraining Data Builds Tamper-Resistant Safeguards into Open-Weight LLMs https://arxiv.org/abs/2508.06601

[2] Best Practices for Biorisk Evaluations on Open-Weight Bio-Foundation Models https://arxiv.org/abs/2510.27629

* Will the pre-trained model be safe against harmful fine-tuning **if we are able to conduct on it a pretty good safety alignment in post-training stage**?

Since a few study discover the harmful fine-tuning risk, e.g., [3-6], there are a rising group of research aim to propose better safety alignment solution that make the model vaccinated against subseqent harmful fine-tuning (e.g., Vaccine[7], RepNoise[8] , TAR[9], Booster[10] and more). While these solutions we are able to strenghten the safety alignment such that it will **refuse to comply with the harmful request** even after harmful fine-tuning. However, several research (e.g., [11])  show that such new solutions are still not robust enough, as they are vulnerabe when the fine-tuning learning rate is large. Indeed, you can always overwrite the safety alignment with stronger attack.

Driven by the weakness of strenghtened safety alignment solutions,  two subsequent research (CTRAP[12] and SEAM[13] ) explore another direction, which is to embed a collapse trap during safety alignment to **make the model collapse if harmful fine-tunig is detected**. I personnally think the collapse trap direction represented by CTRAP and SEAM is the most promising direction to eliminate the fine-tuning risk. Basically, they embed a "shut-down button" into the openweight LLMs and will be activated if the safety alignment is compromised (or  the harmful knowledge is actiated) by harmful fine-tuning.

**[Harmful fine-tuning attack]**

[3] Shadow Alignment: The Ease of Subverting Safely-Aligned Language Models https://arxiv.org/abs/2310.02949

[4] Fine-tuning aligned language models compromises safety, even when users do not intend to! https://arxiv.org/abs/2310.03693

[5] Immunization against harmful fine-tuning attacks https://arxiv.org/abs/2402.16382

[6] Harmful Fine-tuning Attacks and Defenses for Large Language Models: A Survey https://arxiv.org/abs/2409.18169

**[Safety alignment stage defense solution]**

[7] Vaccine: Perturbation-aware alignment for large language model aginst harmful fine-tuning  https://arxiv.org/abs/2402.01109

[8] Representation noising effectively prevents harmful fine-tuning on LLMs  https://arxiv.org/abs/2405.14577

[9] Tamper-Resistant Safeguards for Open-Weight LLMs https://arxiv.org/abs/2408.00761

[10] Booster: Tackling harmful fine-tuning for large language models via attenuating harmful perturbation https://arxiv.org/abs/2409.01586



[11] On Evaluating the Durability of Safeguards for Open-Weight LLMs https://arxiv.org/abs/2412.07097

**[Fine-tuning Collapse Trap]**


[12] CTRAP: Embedding Collapse Trap to Safeguard Large Language Models from Harmful Fine-Tuning https://arxiv.org/abs/2505.16559

[13] Self-Destructive Language Model https://arxiv.org/abs/2505.12186

I think both directions (pre-training stage and safety alignment) that try to mitigate the harmful fine-tuning risk are under significant challenge. This paper should be valuable in terms of pointing out furture research direction and establishing a benchmark for measuring the internal risk of open-weight model.

Best,

Tiansheng Huang

https://huangtiansheng.github.io/

---

### Meta-Review · Area_Chair_eiSn · 2026-01-07

**Summary:**

The main concerns affecting the decision:
- novelty/positioning: the term “malicious fine-tuning (MFT)” is not new, and the framing novelty is over-claimed, with insufficient engagement with prior work. (LjZ5, vNiZ & Trkd)
- experiment details: missing details of the RL/SFT procedures, model weights, hyperparameter choices across models, due to “responsible disclosure”. (Trkd & vNiZ)
- threat model and baselines: missing studies on inconsistent inclusion of open-weight baselines (esp. in cyber), and the adversary has no access to the internal “helpful-only o3”. (Trkd)
- interpretation of results: narrative downplays evidence that gpt-oss (and especially MFT gpt-oss) may raise the open-weight bio frontier, plus a factual/date inconsistency regarding comparison coverage. (YYD1)

**Reviewer Concerns:**

Concerns likely addressed by the rebuttal:
- clarifications/typos/figure inconsistencies: fixed in the revision (QK8y)
- novelty/positioning (MFT): the authors look to reframe as “maximal capabilities elicitation” (LjZ5, vNiZ & Trkd)
- experiment details: partially addressed under responsible disclosure (vNiZ, Trkd, LjZ5)
- threat model and baselines: partially addressed (Trkd, LjZ5)

Concerns still outstanding (or only partially addressed):
- experiment details: without additional methodological/experimential disclosure, confidence and reproducibility concerns remain (vNiZ, Trkd, LjZ5)
- access to the internal model: without additional disclosure/classification, the attacker has no access to the internal "helpful-only o3" model (Trkd)

**Reviewer Scores:**

- LjZ5 (original score 4): likely unchanged or small upward shift
- YYD1 (original score 6): likely unchanged;
- QK8y (original score 8): likely unchanged (already very supportive)
- vNiZ (original score 4): likely small upward shift
- Trkd (original score 6): likely mostly unchanged

---

### Decision · Program_Chairs · 2026-01-26

Accept (Poster)